# Economic and Energy Analysis of Building Retrofitting Using Internal Insulations

**Małgorzata Basińska** [1],*, **Dobrosława Kaczorek** [2] and **Halina Koczyk** [1]

[1] Institute of Environmental Engineering and Building Installations, Faculty of Environmental Engineering and Energy, Poznan University of Technology, 4 Berdychowo Str., 61-131 Poznań, Poland; halina.koczyk@put.poznan.pl

[2] Thermal Physics, Acoustics and Environment Department, Building Research Institute (ITB), 1 Filtrowa Str., 00-611 Warsaw, Poland; d.kaczorek@itb.pl

\* Correspondence: malgorzata.basinska@put.poznan.pl

**Abstract:** The energy-saving requirements for most buildings focus on improving the insulation and airtightness of a building's envelope. In this paper, the authors have investigated the effect of additional internal insulation on energy consumption for heating and cooling in a residential building. Energy performance analyses were conducted for buildings with four internal thermal insulation systems in three locations using the WUFI Plus software. The Global Cost Method and Simply Pay Back Time have been used to assess and compare the economic viability of the retrofit systems. The results show that, in relation to energy, retrofitting with internal wall insulation can be an alternative to traditional external insulation. The assessment of internal insulation for low-energy buildings, however, cannot be conducted based on economic criteria. The usual approach of Simply Pay Back Time has exceptionally long payback time, which is unacceptable. In turn, the Global Cost Method, can only be used to compare the applied materials. With high investment costs, thermo-modernization improvements do not contribute to significant savings. The conditions of thermal comfort and the analysis of temperature and steam pressure play a decisive role in assessing this type of solution.

**Keywords:** envelope design standards; energy savings; internal wall insulation; moisture effects

## 1. Introduction

Reducing energy consumption in existing buildings has now become one of the most critical challenges in building construction, not only in cold climate region [1,2]. Activities undertaken in the area of building's retrofitting focus mainly on improving the thermal insulation of external walls. The amount of energy required for heating and cooling of buildings is primarily determined by the thermal parameters of external walls and windows, which captures up to 25–30% of the total energy loss in a building [3]. Depending on the method of installing the windows in the external wall, the share of losses related to the occurrence of a thermal bridge may be variable. In low-energy buildings, it should not be higher than 5% of the heat transfer loss. Therefore, there is an urgent need to improve the thermal performance of external walls by thermal insulation in order to improve the energy performance of an existing building and to achieve an increase in energy efficiency after retrofitting.

Due to the easier installation, better protection of the wall against weather conditions and the more effective elimination of thermal bridges, the external thermal insulation systems are commonly used solutions [4]. There are some cases where various conditions (technical, administrative, historical) prevent the use of the external systems in buildings. Then the internal insulation becomes the only available solution that can be used. There are many types of internal insulation systems on the market that meet the requirements of high energy efficiency standards [5,6]. However, it should be noticed that internal insulation

may hold many potential threats to the durability of building materials that have been investigated, including damage due to frostbite, interstitial condensation and deterioration of wall-embedded elements, such as the wooden ends of beams [7]. Unfortunately, it is difficult to avoid problems related to the thermal bridges by insulating from the inside [8]. A detailed analysis of the influence of thermal bridges such as floor structure and geometry around windows on the effectiveness of various internal insulation systems in 2D and 3D simulations was presented by Marincioni et al. [9] and Odgaard et al. [10].

The choice type of internal insulation should not only depend on the potential energy savings but also on other additional goals [11,12]. It is necessary to assess the installation possibilities offered by the existing structure and choose a solution that will contribute to minimizing potential problems with moisture and damage resulting from improper use. One of the possible retrofitting materials to use are capillary active materials. A typical representative in this regard is calcium silicate. In many publications, analysis of the hygrothermal risk assessment related to the use of this type of internal insulation in the case of a massive brick wall have been presented [13–16]. The positive influence of this type of insulation of walls has been emphasized by Zhao et al. [17], Zhou et al. [18] or Hamid and Wallenten [19], among others.

The second group of active capillary materials includes biomaterials. For example, the hygrothermal and energy performance of the systems based on soft and hardwood fiberfiberboards were assessed by Kočí et al. [20] using a combination of experimental and computational methods. Wegerer et al. [21] evaluated various interior insulation systems made of wood fiber installed on two demonstration objects. Marincioni and Altamirano-Medina [22] decided to evaluate existing mold growth models for the assessment of mold growth risk in solid masonry walls with internal insulation. Moreover, Jerman et al. [4] argued that hygroscopic properties of internal insulation made of wood fibers are very similar to traditional insulation materials.

The last research concerns the evaluation of the hygrothermal performance of perlite board insulation kits for internal wall retrofitting [23].

The review of the literature, presented above, shows that most studies are concerned with the evaluation of the hygrothermal performance of walls with internal insulation systems. Based on a case study of a building, located in Bamberg, Germany, by Bottino-Leone et al. [24], the comprehensive assessment of hygrothermal, energy and environmental parameters for the adopted retrofit solution was presented for various types of internal insulation (calcium silicate, perlite brick, cellulose, mineral foam, wood fiber, cork). Internal and external thermal insulation systems used in a residential building were compared by Kolaitis et al. [25]. They assessed, inter alia, potential risk for water vapour condensation, annual energy demand for heating and cooling purposes, and from the economic point of view, the payback period. In turn, Piasecki et al. [26] assessed the impact of cellar spaces retrofitting with internal insulation system based on calcium silicate mineral boards in a historical building on the internal environment, using indoor environmental quality index (IEQ).

Only a few pieces of research have analyzed the economic benefit of energy retrofitting with internal insulation [27–30]. In the first three studies, the analysis concerned high-performing vacuum insulation panels, while a recent study looked at different types of insulation, including inorganic natural insulations, such as perlite and microporous calcium silicate. These solutions have been applied to a traditional stone masonry building in Italy using the "Cost optimal" methodology, and the most cost-effective internal insulation system solution was then selected.

For the assessment of the cost-effectiveness of a retrofit solution, several economic analysis methods are used. These methods include: annual operational energy cost savings [31–33], capital costs [34], payback period [33–35] life cycle cost [34,35]; and net present value (also referred to as "global cost") [31,32,35–37].

Building on the aforementioned works, this paper aimed to investigate the energy and economic performance of internal wall insulation systems with organic materials

(wood fiberboards), and inorganic ones (perlite boards and microporous calcium silicate) in Poland's climate zone. We analyzed how moisture flow included in the building's energy balance affects energy consumption, as well as which climate and insulation thickness was a better solution from an economic point of view. In addition, we evaluated how the choice of various methods of economic evaluation of the performed retrofitting affects the profitability of using the analyzed internal insulation.

In order to achieve the above goal, the work was divided into individual stages, which included: defining a building model with the determination of all indirect variables, determining the energy consumption based on numerical simulations, assessing investment costs for individual insulation systems and assessing operating costs, conducting the economic analysis based on the Global Cost Method (GCM) and Simple Pay Back Time (SPBT).

## 2. Methods

A dynamic simulation method with use of the Building Energy Simulation Software—WUFI Plus [38] was used to assess the energy performance of a building. Such software has been experimentally validated by Antretter et al. [39] for a simulation period of one year. The use of WUFI Plus software makes it possible to include, in the modelling, the simultaneous transport of heat and moisture for various building elements (internal and external walls, ceilings and floors). A detailed moisture flow model has been described by Künzel [40]. The algorithm can be used to determine energy consumption for both heating and cooling purposes. In order to perform the energy assessment of a building, it is necessary to provide the information about thermal and humidity properties of building materials used in the partitions, information about location of the building and climatic data, and information about technical systems, along with profile of use.

### 2.1. Simulation Method

An energy simulation of various retrofitting configurations was conducted. The purpose of the simulation analysis was to assess the energy consumption in the selected rooms of a building, taking and not taking the moisture flow into account. The variability of the type of insulation material, its thickness, location in the structure of buildings and two different climate zones were analyzed. In simulations a window with an area of 2.9 m$^2$ and a thermal conductivity of U = 1.1 W/(m$^2$·K) were assumed for each of the analyzed variants. The linear heat transfer coefficient at the interface of frame wall was assumed by $\psi$ = 0.025 W/(m·K). In total, 156 different cases were analyzed using different combinations of: two different climatic zones, three room locations in the building structure, four types of insulation material and three levels of thickness for each material. The method used was based on hourly balances. It included heat transfer losses, heat losses through infiltration and ventilation, and solar and internal heat gains.

One of the main parameters required and used in building simulations is weather data. The climate data used in this simulation are based on the weather data from Warsaw and Cracow. Warsaw is situated at 52.13° north latitude and 21.00° east longitude, Cracow 50.03° north latitude and 19.56° east longitude, in a moderate, warm, transient climate zone. The average temperature in Warsaw is: annual—8.06 °C, in August—16.6 °C, in January——1.2 °C, in turn in Cracow, 8.28 °C, 17.5 °C, and −1.3 °C, respectively.

### 2.2. The Economic Analysis

The economic analysis used the Global Cost Method (GCM) and Simply Pay Back Time (SPBT). In both methods, to determine the value of individual assessment indicators, it is necessary to provide information about investment costs of a given retrofitting improvement and energy savings achieved.

In the dynamic analysis (GCM), we use the fluctuating value of money over time. This variation is described by the value of the discount rate, which enables discounting, i.e., converting the future value of capital to its present value.

In the analyses, in order to determine the operating costs, the discount rate was taken into account at the level of 2.15. This value results from the analysis of the monetary market in Poland. Depending on the adopted value resulting from the economic situation in the country, the obtained results will differ from each other [41].

### 2.2.1. Methodology for the Calculation of the Global Cost

The simplified global cost method used in the analysis takes into account a sum of the initial investment costs of the analyzed individual retrofitting improvements and the discounted annual operating costs, including energy during the calculation period, minus the residual value of each of the components considered. For a residential building, the calculations were conducted for a 30-year period, assuming a building's lifespan of 50 years [42,43].

All algorithms in the working documents of the recast Energy Performance of Buildings Directive (EPBD) [44] are contained. The basic formula on the global cost value is presented below (1) [43]:

$$C_G(\tau) = C_{in,inv} + \sum_{j=1}^{j_x} \left[ \sum_{i=1}^{\tau} (C_{a,i}(j) \cdot R_d(i)) - V_{f,\tau}(j) \right] \tag{1}$$

where: $C_G(\tau)$, the global cost referred to the starting year; $C_{in,inv}$, initial investment costs; $j$, index of component, –; $j_x$, number of components, –; $\tau$, calculation period, a; $C_{a,i}(j)$, annual costs for component $j$ of the year $i$; $R_d(i)$, discount rate (for a year $i$), –; $V_{f,\tau}(j)$, the final value of component $j$ end of the calculation period $\tau$, considering its lifespan and referred to the starting year (Equation (2)).

$$V_{f,\tau}(j) = C_{in,inv}(j) \cdot \left( 1 + \frac{R_p}{100} \right)^{n_\tau(j) \cdot \tau_n(j)} \cdot \left[ \frac{(n_\tau(j) + 1) \cdot \tau_n(j) - \tau}{\tau_n(j)} \right] \cdot R_d(\tau) \tag{2}$$

where additional: $R_p$, the rate of development of the price for products; $n_\tau(j)$, the total number of replacements of components $j$ throughout the calculation period; $R_d(\tau)$, discount rate at the end of calculation period.

The methodology distinguishes between financial calculations and macroeconomic calculations. In the first case, all taxes are included in the cost. In the case of macroeconomic calculations, as used in our analysis, taxes are not included. Instead, these should be replaced by carbon costs. For gas, the factor of carbon dioxide emissions was assumed as 56 $MgCO_2$/TJ, and for the electricity—0.823 $MgCO_2$/MWh [45,46].

### 2.2.2. Methodology for the Calculation of SPBT

A comparison between thermo-modernization materials to initial/base construction was conducted using the Simple Pay Back Time (SPBT) [47]. This defines the time needed for reimbursement of the investment costs borne during the execution of a given investment. It provides the preliminary assessment of thermo-insulation performance, giving the explanatory assessment of asset freeze time, assuming fixed energy costs and disregarding the effects of inflation

The analysis for the walls by means of SPBT was performed using the following formula (3):

$$SPBT = \frac{N}{\sum \Delta O_a} \tag{3}$$

where $N$ is the planned cost of the works connected to decreasing the losses resulting from heat transmittance for the total area of the chosen partition, $\Delta O_a$ is the annual saving on the energy costs resulting from the application of thermo-insulation improvement.

In the assessment of the applied technical solution, reference was made to the value of energy demand for the heating period for a room without additional thermal insulation. Simulation analyses were performed using the WUFI Plus software

### 2.3. Energy Cost

For the cost analysis, energy cost data from the Polish database for the 2020 year was used, assuming for [48]:

- natural gas price, net cost: 0.167 PLN/kWh;
- electric energy price, net cost: 0.52 PLN/kWh.

## 3. Case Study

For the purposes of the analysis, a hypothetical building model was defined—a room located in various locations in the building block (Figure 1). In this way, three models with a different shape factor S/V (S—the external surface of the room, V—its heated volume) were considered. First, the climatic conditions and parameters of the room (S/V) were defined and the energy demand for heating and cooling was determined for the basic variants without additional internal insulation. In the next stage the properties and thicknesses of the materials used as internal insulation systems were defined and the heating/cooling demand was assessed.

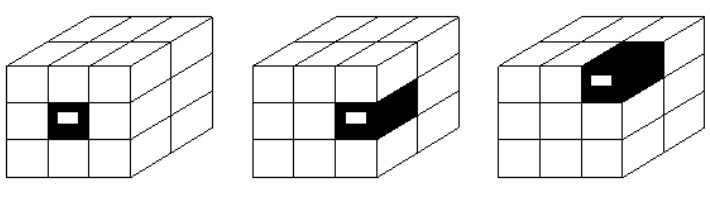

| Number of external walls | unit | 1 | 2 | 3 |
|---|---|---|---|---|
| S | m$^2$ | 14.0 | 34.6 | 64.7 |
| V | m$^3$ | 66.6 | 66.6 | 66.6 |
| S/V | 1/m | 0.21 | 0.52 | 0.97 |

**Figure 1.** Different S/V ratio of the analyzed room.

### 3.1. Reference Room

It was assumed that the tested room with the following dimensions: length—4.0 m, depth—5.5 m, height—2.8 m, is a family room, with a window area of 2.9 m$^2$, facing south.

The thermal properties of a building's exterior base wall were adopted in accordance with the heat protection requirements of buildings in Poland for 2020 [49]. The external base wall was made of two layers: solid brick 25 cm thick and insulation. For external insulation, 15 cm thick mineral wool (MW variant) and 11 cm thick extruded polystyrene (EPS variant) were optionally used, obtaining the same thermal transmittance for the base wall—0.22 W/(m$^2$·K). The applied retrofit solutions were aimed at improving the insulation of an external wall to meet the requirements in force in Poland from 2021 onwards through the use of additional thermal insulation mounted on the inside of the room. The analysis did not take into account the improvement of thermal insulation of other building partitions.

Internal heat gains were estimated by the analysis of the room use profile. It was assumed that two adults and one child would use the room. Different occupancy schedules for weekdays and weekends were adopted (Table 1). The air change was assumed to be 0.5 changes per hour (ACH).

**Table 1.** Occupancy schedule with internal heat gains.

| Occupancy Hours | Weekdays | | | | Weekend | | | |
|---|---|---|---|---|---|---|---|---|
| | Heat Conv. | Heat Radiant. | Moisture | $CO_2$ | Heat Conv. | Heat Radiant. | Moisture | $CO_2$ |
| | W | W | g/h | g/h | W | W | g/h | g/h |
| 6.00–7.00 | 80 | 41 | 59 | 36.3 | – | – | – | – |
| 7.00–8.00 | 220 | 112 | 168 | 106.4 | – | – | – | – |
| 8.00–10.00 | – | – | – | – | 80 | 41 | 59 | 36.3 |
| 10.00–16.00 | – | – | – | – | 220 | 112 | 168 | 106.4 |
| 18.00–20.00 | 220 | 112 | 168 | 106.4 | 220 | 112 | 168 | 106.4 |
| 20.00–22.00 | 160 | 82 | 118 | 72.6 | 160 | 82 | 118 | 72.6 |

*3.2. The Analysed Variants*

Upon calculating and analyzing the energy consumption for heating and cooling purposes for the base version of the building only with external insulation, the decision was made to apply additional internal insulation to further reduce energy consumption. The selection of the appropriate modernization technology depends on many factors, such as: the type of the external wall, location, size and method of building operation and HVAC systems solutions. It is, nevertheless, not always possible for all measures at the same time, besides, the reduction of energy consumption in a building is not the sum of the measures used, but the effect of their mutual interaction [50]. If it is not possible to carry out a comprehensive modernization, it is possible to implement individual solutions in stages, but each new solution should always be carefully analyzed.

3.2.1. Internal Retrofitting Materials—Properties

For the three previously selected locations of a modular room (with different S/V values) in the building structure, four different energy-saving insulating system solutions were used. The selected insulation materials were analyzed with commercial thicknesses from 4 to 12 cm depending on their type. The configuration of added layers and the properties of the main internal retrofitting materials were presented in Table 2 and the resulting values of the thermal transmittance depending on the type and thickness of the additional thermal insulation used in Table 3 were presented.

**Table 2.** Properties of the main internal retrofitting materials for external wall.

| Material Layers | Thermal Conductivity (W/(m·K)) | Heat Capacity (J/(kg·K)) | Density (kg/m³) | μ-Value(–) |
|---|---|---|---|---|
| **A** | | | | |
| rigid wood fiberboard | 0.045 | 2100 | 159 | 4.0 |
| bonding mortar | 0.800 | 850 | 1350 | 16.2 |
| lime plaster | 0.700 | 850 | 1600 | 7.0 |
| **B** | | | | |
| flex wood fiberboard | 0.041 | 2100 | 61 | 3.0 |
| gypsum fiberboard | 0.300 | 1200 | 1153 | 16.0 |
| **C** | | | | |
| adhesive mortar | 0.155 | 850 | 833 | 15.0 |
| microporous CaSi | 0.043 | 850 | 115 | 4.1 |
| adhesive mortar | 0.155 | 850 | 833 | 15.0 |
| lime plaster | 0.700 | 850 | 1600 | 7.0 |
| **D** | | | | |
| bonding mortar | 0.800 | 850 | 1350 | 16.2 |
| perlite board | 0.045 | 850 | 850 | 7.0 |
| mineral plaster | 0.800 | 850 | 190 | 25.0 |

**Table 3.** Final thermal transmittance of the external wall.

| Additional Material | Thickness (m) | U-Value (W/(m$^2$·K)) |
|---|---|---|
| A<br>rigid wood fiberboard | 0.04<br>0.08<br>0.12 | 0.183<br>0.158<br>0.138 |
| B<br>flex wood fiberboard | 0.04<br>0.08<br>0.12 | 0.180<br>0.153<br>0.133 |
| C<br>microporous CaSi | 0.05<br>0.08<br>0.12 | 0.174<br>0.156<br>0.136 |
| D<br>perlite board | 0.05<br>0.08<br>0.12 | 0.176<br>0.158<br>0.138 |

For the calculations, a constant value of material thermal conductivity was used, assuming that the insulating materials used on the inside of the wall are not subjected to temperature fluctuations [28]. The values of the physical properties of insulation materials adopted for calculations were determined in the laboratory of the Building Research Institute as part of the H-house project [51]. The properties of the rest of the materials were adopted based on the WUFI Plus database. The analysis did not include the influence of the thermal bridge on the joints of insulation panels. The authors are aware of the thermal bridging effects due to the air or structural joints between insulation materials, and that the resistance of the bonding layer is important and may influence the final result [52]. Depending on the type of retrofit material, the use of finishing panels or additional plastering was taken into account.

3.2.2. Internal Retrofitting Materials—Investment Cost

The investment cost was estimated for all configurations of analyzed materials. The price analysis was conducted according to the Polish price market. The statements of investment net cost divided into material and labor costs are shown in Table 4.

**Table 4.** Cost of additional thickness of internal retrofitting materials.

| Additional Material | Thickness (m) | Material Cost (PLN/m$^2$) | Labour Cost (PLN/m$^2$) | Total Cost (PLN/m$^2$) |
|---|---|---|---|---|
| A<br>rigid wood fiberboard | 0.04 | 82 | 97 | 180 |
| | 0.08 | 104 | | 201 |
| | 0.12 | 133 | | 230 |
| B<br>flex wood fiberboard | 0.04 | 56 | 117 | 173 |
| | 0.08 | 97 | | 214 |
| | 0.12 | 139 | | 256 |
| C<br>microporous CaSi | 0.05 | 105 | 97 | 202 |
| | 0.08 | 142 | | 239 |
| | 0.12 | 188 | | 285 |
| D<br>perlite board | 0.05 | 118 | 107 | 225 |
| | 0.08 | 153 | | 260 |
| | 0.12 | 216 | | 323 |

### 3.3. Heating and Cooling Systems

The modelling of the technical system of building equipment was not included in the analyses. Temperature for the heating period at the level of 20 °C, and for the cooling period at 26 °C was assumed. A natural gas condensing boiler (used during the winter period) and an air conditioner (used during the summer period) were selected for the simulations. The condensing gas boiler was used as the heat source, assuming its efficiency at the level of 70% (which is the minimum value in Polish law [53]). The air conditioner with the average value of the coefficient of performance (COP) equal to 3 was installed as a source of cooling.

### 3.4. Individual Variants—Abbreviations

The key to the abbreviations used to mark individual analyzed variants is presented in Table 5.

**Table 5.** Abbreviations used to mark individual variants.

| Abbreviation | Description |
| --- | --- |
| EPS | External insulation—expanded polystyrene |
| MW | External insulation—mineral wool |
| K | Climate zone Crakow |
| W | Climate zone Warsaw |
| 1, 2, 3 | Number of external partitions |
| A4, A8, A12 | Internal insulation—rigid wood fiberboard—4, 8, 12 cm thickness |
| B4, B8, B12 | Internal insulation—flex wood fiberboard—4, 8, 12 cm thickness |
| C5, C8, C12 | Internal insulation—microporous CaSi—5, 8, 12 cm thickness |
| D5, D8, D12 | Internal insulation—perlite board—5, 8, 12 cm thickness |

## 4. Results and Discussion

In the paper, the possible variants of the thermal retrofitting of external wall with internal insulation in terms of energy and economy were analyzed. For the evaluation, the algorithms presented in Section 2.2 were used to compare the individual evaluation indicators for each variant.

### 4.1. Energy Analyses

Particular attention in the energy analyses, using the available calculation tools, is paid to the matters related to the energy consumption used for heating and cooling. The energy demand for space heating was calculated for 156 cases for the four insulation materials in two climatic zones. The annual energy consumption of a room was analyzed by changing the insulation of the external partitions. The conducted simulation calculations have allowed for the evaluation of a room's thermal behavior, including the external wall insulated with the thermal insulation from the inside. The authors have drawn particular attention to the fact that it is necessary to calculate the moisture flow, a significant factor when hygroscopic material is applied as thermo-insulation. As the thermal properties of the building's envelope are improving, it is more important to include the cooling seasons into the analyses. Thus, simulation analyses were conducted for both heating and cooling seasons. The length of those periods change depending on the applied thermo-insulation material, the walls' thermal insulation and the position of the room in the building in different climate zones.

#### 4.1.1. Analysis of Annual Heating and Cooling Energy Consumption of Rooms Calculated with or without Moisture

According to Yanga et al. [54], in the energy assessment of a building it is necessary to take into account the moisture and heat transfer through building envelopes and between the building envelope and the indoor environment. Zhang et al. [55] proposed the coupled heat, air and moisture transfer (HAMT) model [40,56,57] as the most appropriate to include

the impact of humidity content on a building's energy consumption during simulation calculations. The HAMT model is available in building simulation tools such as WUFI Plus.

In this study, a building's energy consumption for heating and cooling includes a comparison between two types: with and without the use of a humidity model. The results of our analyses have shown that the percentage difference (regarding the energy consumption without moisture) is greater for the heating season. In all analyzed variants for the climate data regarding Warsaw, this is approximately 6.0% for the heating season and 1.2% for the cooling season. For Cracow, this is 6.9% and 3.1% respectively. The 1.9% higher average percentage difference for the cooling season in Cracow results from a higher 0.9% average temperature for that season. The average consumption during the heating season for type (3) walls is comparable; however, a significant difference occurs in the cooling season. For example, in the case of Warsaw, the annual consumption for heating equals 3435.3 kWh/a, while for Cracow it is 7.7 kWh/a less. In the cooling season that difference reaches 96.0 kWh/a with a value for Warsaw equal to 874.5 kWh/a. This also changes depending on room localization in the building. The higher heat losses (room with three external walls) are less significant than the percentage change of heat consumption. In the case of Warsaw's climate data, for a middle room with one external wall (1) this changes from a minimum value of 6.6% to a maximum of 8.8% for the heating season and from 1.0% to 2.4%—for the cooling season. In the case of room type (2) with two external walls this rises from 4.7% to 6.6% for the heating season and from 0.5% to 2.5% for the cooling season. In case of room type (3) with three external walls, this is from 3.8% to 5.3% for the heating season and from 0.0% to 2.1% for the cooling season, respectively. As per Cracow's climate data, in the case of:

- The middle room, type (1)

  ○ 7.8 to 10.4%—heating season
  ○ 2.0 to 3.2%—cooling season

- The corner room, type (2)

  ○ 5.5 to 12.5%—heating season
  ○ 2.0 to 10.0%—cooling season

- Room type (3)

  ○ 0.0 to 6.1%—heating season
  ○ 2.0 to 8.7%—cooling season

As an example, Figure 2 presents the percentage change of the energy consumption for individual purposes with or without moisture for Warsaw, considering an 8 cm thick thermo-insulation layer.

Depending on the analyzed variant and method applied, the length of a particular season has also changed for: H—heating, C—cooling T—temporal seasons. In our analyses we noticed that the value of energy consumption is higher in cases where moisture was included to the calculations in comparison to analyses where moisture was not included, while shortening the heating season and extending the cooling season. This change is insignificant, and equals, for Warsaw's cold season, to approximately −2.9%, and −4.7% for the warm season. Accordingly, it ranges from 2416 h (room with one external wall, analysis with moisture flow) up to 3065 h (room with three external walls, analysis without moisture flow) for the heating season, and from 794 h (three external walls, analysis with moisture flow) to 1496 h (one external wall, without moisture flow) for the cooling season. The average length of the temporal period is 4612 h. For Cracow, similar dependencies have been noticed. The average difference between the time of the indicated seasons with or without moisture flow is 87.1 h and for the analyses including moisture flow the average duration of the seasons is for:

- The middle room, type (1)

  ○ heating time—2471 h
  ○ cooling time—1375 h

- ○ temporal time—4915 h
- The corner room, type (2)
  - ○ heating time—2774 h
  - ○ cooling time—1248 h
  - ○ temporal time—4738 h
- Room type (3)
  - ○ heating time—3428 h
  - ○ cooling time—779 h
  - ○ temporal time—4554 h.

In terms of the length of the temporal period, when assuming the limit of temperatures at a level of $\theta_H = 20\ ^\circ$C and $\theta_C = 26\ ^\circ$C, the number of hours seems to be surprisingly high; it does however reduce when the heat losses during the heating season increase.

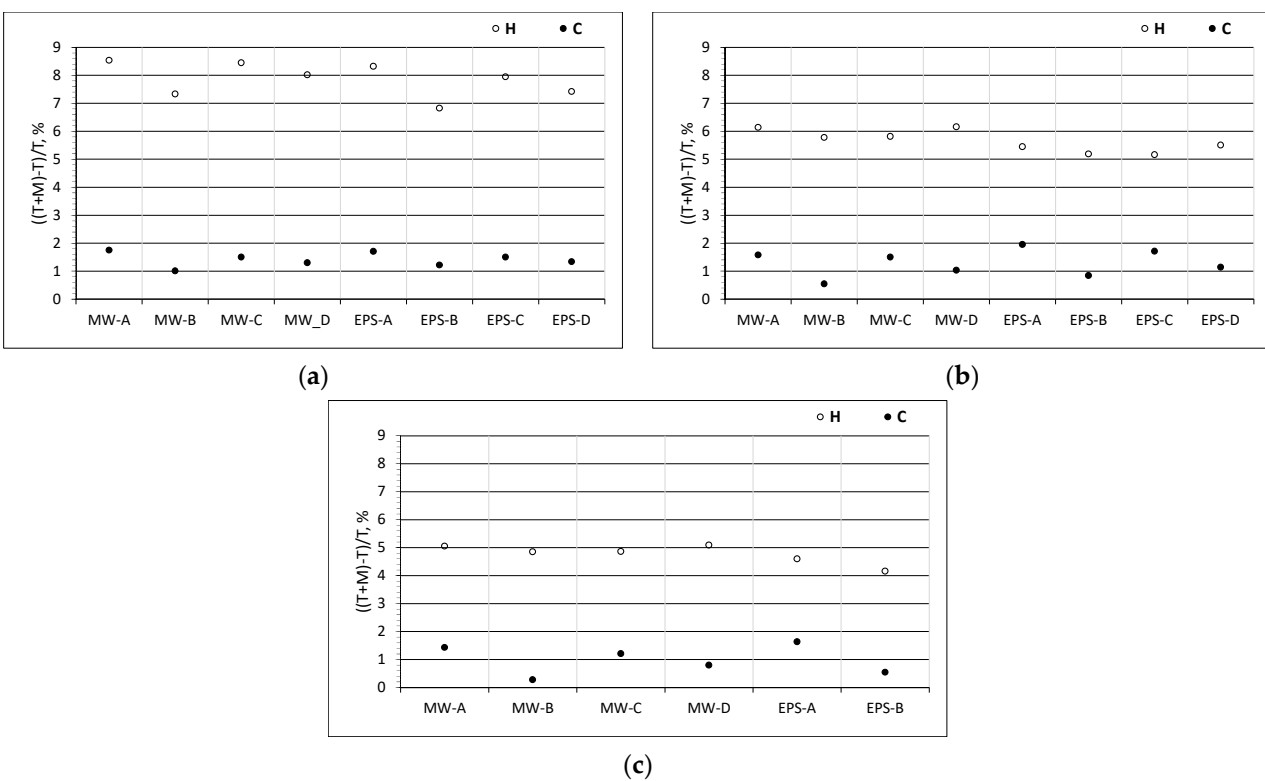

**Figure 2.** Percentage change of energy consumption for heating, ventilation and cooling with or without moisture for climate data from Warsaw and an 8 cm thick thermo-insulation layer, (**a**) one external wall, (**b**) two external walls, (**c**) three external walls.

Based on the above, it can be indicated that when the energy analyses include the moisture flow, a complete picture of the influence of physical phenomena on the energy consumption in the room has been obtained.

4.1.2. Analysis of Annual Heating and Cooling Energy Consumption of Rooms with a Different Type of External Insulation

One of the conditions for fulfilling the building heat protection requirements prescribed in Polish technical requirements is not to exceed the limit value of thermal transmittance for a given wall [47]. There are no special requirements regarding the wall structure and applied thermo-insulation material. As condensation is possible in the wall, it is important to keep the correct sequence of materials placement with their different thickness and properties. In this paper, the effect of the application of different insulation materials on the energy consumption value has been analyzed. The analyses were conducted

under the assumption of a different starting structure of the external wall with the use of MW or EPS. In order to achieve the same values of thermal transmittance of a basic wall (the different values of thermal conductivity for external insulation: MV—$\lambda = 0.04$ W/(m·K), EPS—$\lambda = 0.03$ W/(m·K)) it was necessary to assume a different thickness of external insulation material, and as a result, the area of the external wall was insignificantly changed. The different hygro-thermal properties dictated the materials used. The difference in the annual energy consumption depends on the type of external insulation regarding 1 sqm of the heated area with an adjustable temperature for the calculations including moisture flow and has been lower in comparison to the calculations not including moisture flow. For the external wall without additional thermal insulation, in Warsaw climate data and the heating season were equal to: in room (1)—0.06 kWh/(m²·a) with moisture flow and 0.20 kWh/(m²·a) without moisture flow, for room (2) 0.27 kWh/(m²·a) and 0.06 kWh/(m²·a) respectively, for room (3)—0.28 kWh/(m²·a) and 0.64 kWh/(m²·a) with the values of usable energy as presented in Table 6.

**Table 6.** The value of usable energy for individual purposes.

| Insulation Material of External Wall | Type of Room | Heating (kWh/(m²·a)) | | Cooling (kWh/(m²·a)) Data | |
|---|---|---|---|---|---|
| | | Warsaw | Cracow | Warsaw | Cracow |
| MW | (1) | 28.3 | 26.6 | 14.1 | 13.4 |
| | (2) | 39.6 | 37.7 | 11.1 | 10.7 |
| | (3) | 55.6 | 53.0 | 7.9 | 7.6 |
| EPS | (1) | 28.4 | 26.4 | 14.1 | 13.5 |
| | (2) | 39.8 | 37.9 | 11.1 | 10.7 |
| | (3) | 55.8 | 53.4 | 7.8 | 7.6 |

Where the analyses included moisture flow, despite the significant difference in the water vapor resistance coefficient values—$\mu$ for the external insulation of the external wall (for XPS—$\mu = 50.0$, for MW—$\mu = 1.3$), an insignificant percentage change of the usable energy value for heating purposes was noted (for Warsaw, room (1)—0.35%, (2)—0.50%, (3)—0.36%). For Cracow, the percentage change for individual rooms was at a level of $\pm 0.76\%$. The presented analyses clearly indicate that the type of external insulation (different diffusion resistance of the applied material), while keeping the same value of thermal transmittance of the wall, has an insignificant effect on the change of usable energy value for particular purposes.

4.1.3. Analysis of Annual Heating and Cooling Energy Consumption According to Insulation Type

If the insulation of the wall cannot be placed on its external side, additional thermo-insulation must be installed inside. When using such a type of insulation, moisture flow has a significant influence on the energy consumption value for the particular purposes (Section 4.1.1). Evaluating a given type of internal insulation and assuming only the water vapor resistance coefficient $\mu$ of a given insulation element not including the finishing component, there was lack of correlation between the $\mu$ value and the usable energy for all analyzed thicknesses. The above results from the fact that the type of surface material is significant, as the place where direct contact with moisture from the rooms occurs. Particularly crucial factors are its hygric properties, like water absorption, moisture diffusivity, water vapor resistance and water adsorption isotherms.

When conducting the energy analyses, attention must be drawn to the percentage change of the usable energy consumption for the individual purposes in reference to the usage of the basic wall. Based on the data for Warsaw (calculations with humidity model), the use of a 12 cm thick internal insulation decreases the energy usage for heating purpose by a 10.9% for room type (1), 24.9% for room type (2), and 19.2% for room type (3). This change is greater for material characterized with a lower diffusion resistance coefficient,

internal insulation type B. For the cooling option, the energy consumption for cooling purposes increases, up to 36.1% for room type (1) for internal thermal insulation type B and external thermal insulation made of MW.

For example, external insulation made of MW is presented in Figure 3 for the individual energy consumption for heating purposes. At the additional grid line, the percentage change has been presented in comparison to the wall without additional internal thermal insulation. The chart's first pair of data presents the usable energy value for the wall without the additional internal insulation—reference value (basic wall).

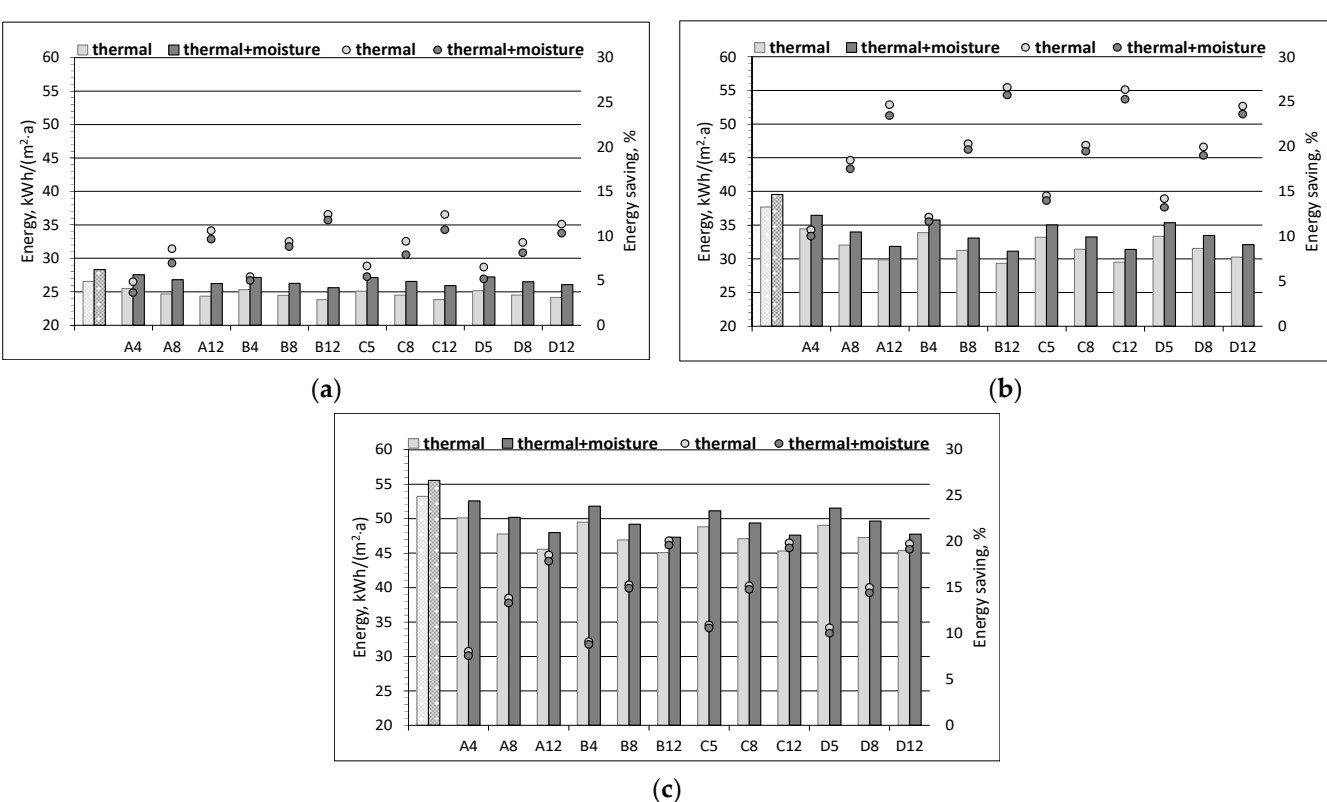

**Figure 3.** The individual usage of energy for heating and ventilation purposes for the analyzed types and thicknesses of internal thermo-insulation for Warsaw and external insulation made of MW, (**a**) one external wall, (**b**) two external walls, (**c**) three external walls.

For each of the analyzed cases, the percentage change is greater than for the analyses, not including moisture flow. The application of additional thermal insulation decreases heat consumption for heating and ventilation purposes by up to 26.1% for the room with two external walls and external insulation made of EPS. For the insulation made of MW, the decrease amounts to 25.7% regarding heat losses without additional internal insulation.

4.1.4. Analysis of Annual Heating and Cooling Energy Consumption of Building According to Climate Data

A factor that has a huge impact on the energy consumption of a building is its climate zone localization. An analysis into the energy consumption depending on the localization in a given climate zone shows a decrease of energy consumption by approximately 6.3% for heating without humidity flow, and by 6.4% for cooling when comparing the climate data of Cracow and Warsaw. The percentage difference is smaller for the analyses including, heat flow and moisture flow, and amounts to 5.5% for the heating season and 4.7% for the cooling season. This difference is reduced when heat losses increase. In an assessment of the building from an energy performance point of view, its localization is of paramount

importance and results from the parameters' changeability and climate processes. The acceptable solutions for one localization do not have to suit another.

### 4.1.5. Final Energy

Final energy has been included in the economic analyses assuming the efficiency of the heating and cooling systems in accordance with Section 3.3. Energy analyses were made, considering the heat and moisture flows. Energy consumption for hot water preparation was not analyzed. For the climatic data of Warsaw, for room type (1), the heating season reduction is from the value of $E_{K,H,0} = 46.0$ to $E_{K,H,n} = 42.9$ kWh/(m²·a), for room type (2) from $E_{K,H,0} = 61.1$ to $E_{K,H,n} = 56.0$ kWh/(m²·a) and for room type (3) from $E_{K,H,0} = 83.1$ to $E_{K,H,n} = 72.2$ kWh/(m²·a). For all cases, the lowest final energy value was for insulation type B.

### *4.2. Economic Analysis*

In the economic analysis, the impact of the variability of the parameters were taken into account in the building's energy evaluation, described with the final energy coefficient, and the value of economic factors were discussed and assessed.

### 4.2.1. Total Energy and SPBT

In order to assess the thermo-modernization improvements conducted in the building, the SPBT method is normally used. An assessment based on SPBT depends on the initial thermal characteristics of the building. Due to the modernization of a low-energy building, a direct application of SPBT may not be a suitable solution. In this paper, thermo-modernization improvements consist of placing the additional internal insulation ($U_{sz,o} = 0.22$ W/(m²·K)) on the well-insulated external wall. For the purpose of assessment, the change of SPBT value of the given technical solution was used concerning the minimum value obtained locally for a given room in the building.

Based on the conducted analyses into SPBT changeability, the highest was noted for material A in room type (1) (Figure 4)—greater curve slope. For others material, the SPBT values do not change with the increase of insulation thickness from 8 cm and above.

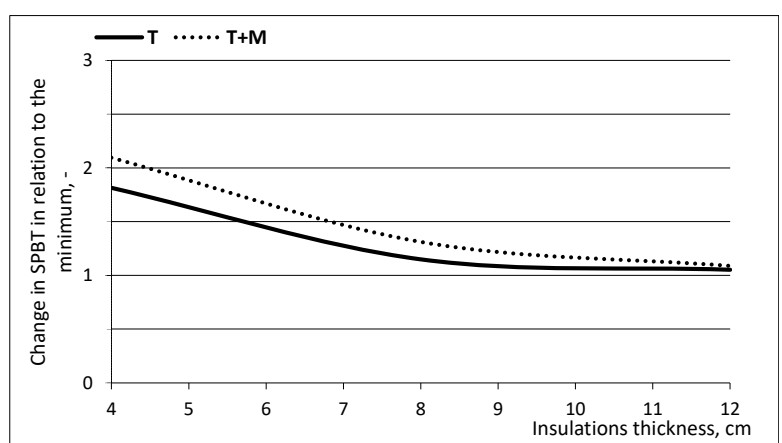

**Figure 4.** Change of SPBT depending on the type and thickness of internal thermo-insulation for the localization of analyzed rooms in the building for the Warsaw climate zone and external insulation made of MW, material A, one external wall.

When designated with the additional moisture flow SPBT shows a higher value by 11% in comparison to the calculations without moisture flow. For materials and their analyzed thickness in room type (1), the minimum value $SPBT_{min} = 133$ years was for material type B12 (thickness of 12 cm). This value is more than acceptable level.

Using SPBT to assess the technical solutions of these insulations in the building is demonstrative (high values of SPBT) and difficult to verify in practice. According to the

authors' opinion, this method may only be applied for comparing the used materials with a given thickness.

An excellent tool for the assessment of thermo-modernization improvements (equally popular) might be the global cost method, and not just for low-energy buildings. Among the advantages of this method are assuming the discount rate in the calculations as well as the value change in real-time of the energy and investment costs.

### 4.2.2. Total Energy and Global Cost

The global cost method has been used in many European countries to determine the requirements of a building's thermal protection. The searched parameter was a primary energy ratio for the building in the minimum function of the global cost value. This paper presents the use of the global cost method algorithm for assessing the proposed solution of the internal thermal insulation of the external wall. In Table 7, for example, the values of the global cost for the analyzed cases of the internal insulation have been presented, for Warsaw's climatic data, external insulation made of MW and final energy designated by taking into account the moisture flow.

**Table 7.** The value of global cost for variants for the Warsaw climatic zone.

| Insulation Material of External Wall | Type of Room | $C_G$, (PLN/m$^2$) | | | | |
|---|---|---|---|---|---|---|
| | | Base | Average | SD | Min | Max |
| MW | (1) | 221 | 293 | 14,6 | 273 | 325 |
| | (2) | 282 | 484 | 40.8 | 430 | 574 |
| | (3) | 373 | 574 | 39.9 | 522 | 662 |

Based on the conducted analyses, it is clear that the global cost increases with the thickness of the insulation material. For all types of insulation materials, in all locations in the building, decreased energy consumption costs, resulting from the applied technical solution, do not counterbalance the discounted investment and environmental costs—$CO_2$. A full picture of the global cost value change will act as a reference to the value of the final energy for the given analyzed case.

As the final energy value for three-room localization in the building is variable, the increase of the global cost due to the change of the final energy for heating, ventilation and cooling purposes has been presented in Figure 5.

The analysis of the global costs method results shows that for the nearly zero energy buildings, the application of additional thermal insulation does not bring any significant savings designated in PLN/m$^2$. This is beyond thermo-modernization profitability. If due to technical reasons, the placing of internal insulation is recommended, we can only compare the used materials in the global cost method.

When the investment costs are high (material 1), the curve slope is the biggest, which denotes the highest increase in global cost. The flattest curve refers to material A, where the lowest percentage change of global cost can be noticed when the insulation material is becoming thicker. In the case of material B, there is no significant difference in percentage change determined with and without moisture flow taken into account. For material B only room localization with type (1) in the building indicates the increase of global costs for the analyses with moisture flow. In the case of material C for all analyzed localizations in the building, the increase of the global cost is lesser for the analyses with moisture flow.

The assessment of internal insulation for low-energy buildings cannot be carried out based on economic criteria. The thermal comfort conditions and analysis of temperature fields and water vapor pressures play a decisive role in the process of evaluation of interior insulation materials.

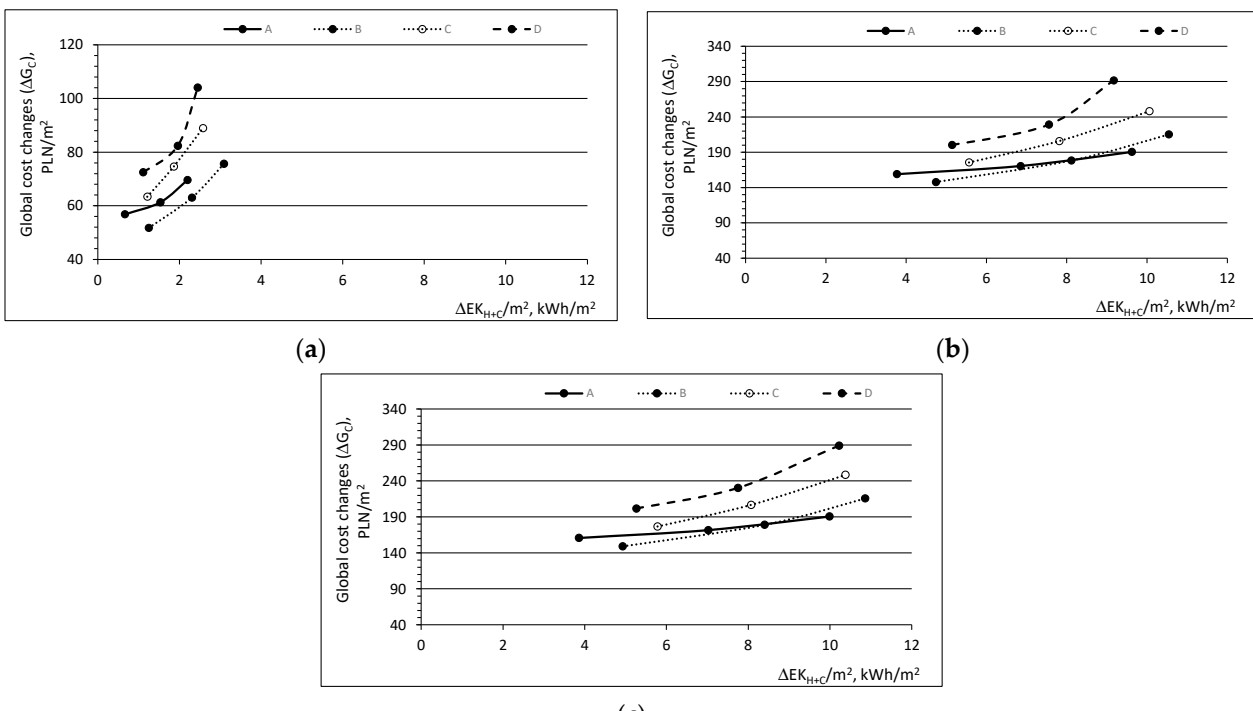

**Figure 5.** Increase of global cost in relation to the change of the value of final energy for the analyzed types and thicknesses of internal thermo-insulation for the Warsaw climatic zone and external insulation made of MW, (**a**) one external wall, (**b**) two external walls, (**c**) three external walls.

## 5. Conclusions

This article presents an energy and economic analysis for the issue of retrofitting internal insulation walls in a room in three different locations in a residential building. For two types of base wall, one with MW as an external insulation and another with EPS, four internal insulation systems were compared: A—rigid wood fiberboard, B—flex wood fiberboard, C—microporous CaSi, D—perlite board. For each case, the energy consumption for heating and cooling was determined for two climatic zones (Warsaw and Cracow) using WUFI Plus simulations, with and without taking moisture flow into account.

Among all analyzed cases, solution B with 12 cm thickness, showed the highest total energy saving. For example, for Warsaw's climatic conditions, including moisture flow the decrease of heating and ventilation demand purposes was at the level of 73.5 kWh/a for room type (1), 224.0 kWh/a for room type (2), and 240.1 kWh/a for room type (3). Without moisture, these values were—72.7 kWh/a for room type (1), 220.6 kWh/a for room type (2), and 235.0 kWh/a for room type (3). In addition, the analyses show that the percentage difference in energy consumption in the calculations with moisture flow for heating is greater than for cooling. For Warsaw, this is 6% for heating and 1.2% for cooling. For Cracow, this is 6.9% and 3.1% respectively. This change corresponds with a shortening of the heating period by 2.9% and an extension of the cooling period by 4.7% for Warsaw and by 3.4% and 8.4% respectively for Cracow. The results concerning the impact of the climate zone on energy consumption showed a reduction of energy consumption for Cracow by 6.3% for heating and 6.4% for cooling without moisture flow and by 5.5% for heating and 4.7% for cooling with.

In the global cost method assessment, variant B with 4 cm insulation turned out to be the most beneficial for all types of rooms.

After the conducted analysis of the results, the following conclusions can be stated:

- all retrofit variants decreased heating energy consumption, but an increase of cooling energy was much lower, and all variants were able to reduce total energy consumption;

- the value of energy consumption is higher in cases where moisture was included to the calculations in comparison to analyses where moisture was not included;
- the higher heat losses (room with three external walls) are less significant than the percentage change of heat consumption;
- when the energy analyses include the moisture flow a complete picture of the influence of physical phenomena on the energy consumption in the room has been obtained;
- the difference in the annual energy consumption depends on type of the external insulation regarding 1 sqm of the heated area with an adjustable temperature for the calculations including moisture flow, and has been lower in comparison to the calculations not including moisture flow;
- evaluating a given type of internal insulation and assuming only the water vapour resistance coefficient μ of a given insulation element not including the finishing component, there was lack of correlation between the μ value and the usable energy for all analysed thicknesses;
- when conducting the energy analyses, attention must be drawn to the percentage change of the usable energy consumption for the individual purposes in reference to the usage of the basic wall.

Furthermore, the results show that, from an economic point of view, the internal wall insulation systems for low-energy buildings become less effective. This depends on the fact that for buildings characterized by a low value of thermal transmittance at the level of $U = 0.22 \text{ W}/(\text{m}^2 \cdot \text{K})$, additional thickness of insulation does not convert into large energy savings, as there are relatively high investment and energy costs. Moreover, the application of the method, considering the discount rate in calculations and the change in the value of investment and energy costs over time does not lead to significant savings. It can, therefore, be concluded that the retrofitting of buildings with low-energy consumption using internal wall insulation cannot be carried out only based on economic criteria.

**Author Contributions:** Conceptualization, M.B., D.K. and H.K.; data curation, M.B.; formal analysis, M.B., D.K. and H.K.; investigation, M.B., D.K. and H.K.; methodology, M.B. and D.K.; resources, D.K.; supervision, H.K.; visualization, M.B. and D.K.; writing—original draft, M.B. and D.K.; writing—review and editing, M.B. and D.K. All authors have read and agreed to the published version of the manuscript.

**Funding:** This research received no external funding.

**Institutional Review Board Statement:** Not applicable.

**Informed Consent Statement:** Not applicable.

**Conflicts of Interest:** The authors declare no conflict of interest.

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
