# Peer review of "Economic and Energy Analysis of Building Retrofitting Using Internal Insulations"

_energies, doi:10.3390/en14092446_

Round 1
Reviewer 1 Report
.
Reviewer 2 Report
Please check the English language, for instance,
- line 14 "In a large part of building requirements for energy saving are concentrated on improving the level of the insulation and airtightness of a building’s envelope, especially"
- line 74 "by Bemberg, Germany" should be "in Bamberg"
- line 110 "2. materials"
- line 126 "taking and not taking into account"
- line 157 explain abbreviation "EPBD"
- line 160-163, unnecessary to present PLN in each variable
- line 180, "per given years PLN/year".
Please check the references in red color.
In Equation (1), shouldn't the final value Vf,(j) also be discounted? Also, please check the terminology for 'final value' and its reference [42].
In Equation (2), what about the discount rate? Also check its reference [45].
Reviewer 3 Report
The paper deals with an interesting topic in a field that is experiencing a coinsiderable development in recent years. The conclusions of the study are relevant. However, the paper need further development of the introduction and has several style issues, which should be amended for publication. A professional or native-English speaker revision is strongly advised, as the general readability of the paper should be improved.
In the following lines, a detailed commentary upon some aspects is provided:
ABSTRACT. Please rewrite the first sentence, as it is very confusing.
INTRODUCTION.
Lines 36-37. Please include further explanation regarding the influence of other elements (thermal bridges, windows, etc.) on the overall performance; this is crucial for the success of an either internal or external insulation.
MATERIALS.
Please provide a graphical description of the studied insulating solutions.
3.1. Please provide information regarding the window material and thermal bridge due to window frames, in order to make the simulation repeatable.
Conclusions. Please include a conclusion comparing the different solutions and materials in a explicit way, both in termf of minimum cost and of maximum efficiency. The differences between all parameters in the study should be covered in the discussion and in the conclusion section.
Round 2
Reviewer 2 Report
The reviewer has no further comments.
Reviewer 3 Report
The authors have properly amended the indicated issues and the paper has been adequately improved.